# An Infrastructure Management Humanistic Approach for Smart Cities Development, Evolution, and Sustainability †

Carlos M. Chang [1,*], Gianine Tejada Salinas [2], Teresa Salinas Gamero [3], Stella Schroeder [4,5], Mario A. Vélez Canchanya [6] and Syeda Lamiya Mahnaz [1]

[1] Engineering Center, Department of Civil and Environmental Engineering, College of Engineering and Computing, Florida International University, Miami, FL 33174, USA; smahn002@fiu.edu
[2] College of Engineering, Universidad Ricardo Palma, Lima 15039, Peru; gianine.tejada@urp.edu.pe
[3] Instituto del Pensamiento Complejo Edgar Morin, Universidad Ricardo Palma, Lima 15039, Peru; teresa.salinas@urp.edu.pe
[4] Department of Architecture and Territory, College of Engineering, Universidad de Piura, Piura 20009, Peru; stella.schroeder@gmx.de
[5] Department of Construction and Design, College of Architecture, Universidad de Bio-Bio, Concepcion 4130000, Chile
[6] MAVC, Contratistas Generales E.I.R.L, Lima 15084, Peru; mariovelez@mavcproyectos.com
[*] Correspondence: cachang@fiu.edu
[†] This paper combines two presentations delivered at the 5th International Conference on Transportation Infrastructures (V ICTI) in Lima, Peru in August 2022. It has been expanded and selected for publication in this journal.

**Abstract:** Over the next decades, people will continue moving to urban areas all over the world, increasing infrastructure needs to satisfy economic, environmental, and social demands. The connection between civil urban infrastructure and smart cities is strong due to the common goal of fulfilling public service demands. Infrastructure management contributes to the development, evolution, and sustainability of smart cities. The main problem with traditional approaches to the development, evolution, and sustainability of smart cities is the lack of a holistic, integrated vision of infrastructure management. The main objective of this research is to introduce an infrastructure management humanistic approach with a smart city conceptual model that also considers an educational perspective. A mixed research methodology that combines quantitative and qualitative approaches was used, applying inductive-deductive tools. The paper concludes with the development of an infrastructure management framework for smart cities with five dimensions: (1) Environmental, (2) financial-economic, (3) political-governance, (4) social-people, and (5) technological. Two case studies for the cities of Lima and Piura in Perú illustrate how to incorporate this framework into practice. The research products are relevant because they foster an inclusive better quality of life for all citizens by preserving civil infrastructure systems.

**Keywords:** civil infrastructure systems; management humanistic approach; smart city; sustainability; 5-dimensional model; BIM-CIM; digital twin; Peru

## 1. Introduction

Infrastructure, in the broadest sense, is defined as "The system of public works of a country, state, or region; also: the resources (as personnel, buildings, or equipment) required for an activity" [1]. Civil infrastructure systems have a major impact on the economic development of a nation, and public demand for infrastructure facilities is constantly increasing. In this context, the implementation of effective infrastructure management practices is critical to fulfilling public service demands. "Infrastructure asset management includes the systematic, coordinated planning and programming of investments or expenditures, as well as the design, construction, maintenance, operation, and in-service

evaluation of physical infrastructures and associated facilities. It is a broad process, covering those activities involved in providing and maintaining infrastructure at a level of service acceptable to the public, intended users, or owners" [2].

The concept of a smart city originated in the 1960s and 1970s with the use of databases and aerial photography to collect the data required for cluster analysis in urban planning [3]. The aim was to make well-informed decisions to effectively allocate resources in order to reduce poverty by fostering social and economic growth. As the concept evolved from theory to practice, smart cities sought municipal solutions using new technology and innovative engineering approaches. At present, the concept has been expanded with the development of holistic urban models that involve public enabling, social inclusion, and community engagement in the planning and management process.

From a broader perspective, a city's smartness is determined by a set of characteristics: The level of development of urban infrastructure, technological tools, environmental initiatives, functional public transportation, assertive and progressive city plans, and the ability of people to live and work using urban resources [3]. For a city to become smart, it must also be resilient to effectively deal with rapid climate change and global warming, deforestation due to growing urban areas, limited power generation supplies, increasing demands for manufacturing goods and food production, and the emergence of new modes of transportation.

The United Nations (UN) states that the world's urban population will reach 68% by 2050, increasing energy consumption and emissions with every passing year. Over the next three decades, 2.5 billion people will move to urban areas all over the world, causing substantial impacts on economic, environmental, and social demands [4]. In September 2015, the Sustainable Development Goals (SDGs) were proposed by the United Nations, and 193 countries agreed to seventeen goals aiming to eradicate poverty, protect the planet, and ensure peace and global prosperity by 2030. Goal 9 is "Innovation and Infrastructure" and goal 11 of the SDGs focuses on "Sustainable Cities and Communities". Therefore, it is acknowledged that without a significant transformation of construction and infrastructure management practices, it would be impossible to achieve sustainable development [5].

Furthermore, the National Academy of Engineering (NAE) mentions that "Restore and improve infrastructure" is one of the 14 Grand Challenges for Engineering in the 21st Century [6]. The challenge is not only in the engineering and management of the technical aspects. A sustainable environment and scalable development of the economy affect the quality of life and overall satisfaction of the people in cities. The smart city concept aims to improve the quality of life of citizens by sustaining a resilient urban environment amidst global challenges and local concerns [7,8].

It is important to emphasize that the main priority of smart city initiatives is to fulfill the needs of all citizens, and this is only possible with accessible civil infrastructure systems. The authors believe that this is what makes a city inclusive. An inclusive city should provide a basic supply of public services, quality health and education, decent housing, an efficient transport network, and access to public space and leisure for all citizens.

## 1.1. Research Motivation and Context

Infrastructure management practices can contribute to the development, evolution, and sustainability of smart cities. The connection between civil urban infrastructure and smart cities is strong due to the common goal of fulfilling public service demands (e.g., communications, electricity, housing, transportation, and water). The research motivation is to contribute to this development by providing an infrastructure management framework that considers humanistic and educational perspectives.

This paper presents an infrastructure management humanistic approach developed by the authors for smart cities in the context of the UN Sustainable Development Goals. It was prepared in response to the call for selected topics for the 5th International Conference on Transportation Infrastructures Conference (V ICTI 2022) in Lima, Perú, in August 2023. Therefore, the case studies in this paper refer to the cities of Lima and Piura in Peru.

"Peru is located in South America with "a population of over 32 million, and its capital and largest city is Lima. At 1,285,216 km$^2$ (496,225 sq mi), Peru is the 19th largest country in the world and the third largest in South America" [9]. At present, Peru is undergoing a governmental restructuring in recognition of the importance of infrastructure systems and the role of education in the development of a nation. It is the aim of the authors that this paper will provide guidance to foster a proactive dialogue across the governmental, business, academic, and general public sectors. It is also expected that the conceptual approach for smart cities, the infrastructure management framework, and lessons learned from case study experiences be adapted to other cities.

### 1.2. Research Problem

Local governments need to address increasing economic, environmental, and social demands. Nowadays, technology brings new tools to collect and analyze data for decision making, but at the same time, challenges arise with the digital revolution, leading to significant changes in society that require innovative management approaches. In addition, a smart city should be inclusive and foster respect for its historical urban infrastructure heritage by connecting people to facilitate the integration of cultural values in the community as prone by governmental policies that promote higher standards of living.

The research problem is posed due to the nature of smart cities and the lack of an infrastructure management vision of traditional approaches. The concept of smart cities is interdisciplinary in nature and demands a new way of managing the city with the help of advanced knowledge and technologies without losing sight of human needs. Within this concept, it is necessary to model the various scenarios that can occur in the city, identify the causes of the crisis, analyze them, and seek the best solutions.

The problem is that traditional management approaches have been focused on a single aspect of smart cities (e.g., technology) when there is a need to integrate multiple dimensions to address infrastructure problems. In this sense, the linear, disjointed, and fragmented approaches to managing smart cities do not work. New approaches are required to address the complex dynamics of cities, providing citizens and local governments with a greater capacity to successfully face infrastructure challenges.

### 1.3. Research Objective

The main objective of the research is to develop an infrastructure management framework that considers the multi-dimensional nature of smart cities while understanding humanistic and educational perspectives in the solution of urban infrastructure problems. In this context, the research objective seeks to answer the following question: Is it possible to develop an infrastructure management framework for smart cities to capture the multiple dimensions and complex evolving dynamics of urban infrastructure systems that affect the quality of life of the citizens?

It is claimed by the authors that cities are complex and infrastructure problems should be addressed following a humanistic-centered approach supported by education and modern technologies for the efficient interaction of education, health, transportation, public safety, energy, and building management, among other subsystems.

### 1.4. Research Significance and Relevance

The significance of this paper is its contribution to an infrastructure management framework that integrates a holistic, multidimensional conceptual approach able to foster the development, evolution, and sustainability of smart cities. The approach is centered on preserving urban civil infrastructure to enhance the quality of life of the citizens. In this sense, the authors conceive that a smart city is one that has developed civil infrastructure facilities as well as organizational and operational capacity to meet the needs of its citizens. The conceptual model proposed by the authors integrates the five dimensions (5D) of a smart city: environmental, financial-economic, political governance, social people, and technological into infrastructure management practices.

Infrastructure management practices are crucial for the development, evolution, and sustainability of smart cities. In this sense, it is recognized that the application of new technologies enables deliberate communication and automated data collection from citizens that are useful in the management of urban infrastructure [10]. A successful example of the application of management practices for smart cities is the Milton Keynes (MK) smart city initiative in the UK, operated by the Open University (OU). This UK initiative focused on improving infrastructure system network domains such as water efficiency, energy usage, and transportation. They established an MK Data Hub related to energy, water, and transport infrastructure and gathered open data from infrastructure networks, sensor data, and social media platforms [11]. Another example is the Smart Mobility 2020 Initiative in Singapore, which developed an intelligent transportation system to foster a connected and interactive land transport community [12].

In addition to these examples, the research examined worldwide initiatives for smart cities proposing both cutting-edge technologies and digital literacy as implanted by smart education. Smart education promotes the adoption of sustainable local government policies supported by digital platforms that seek public opinion, fostering critical thinking to address economic, environmental, and social urban infrastructure challenges. This research is relevant because it encourages the proactive participation of the citizens in the planning and management of smart city infrastructures.

*1.5. Organization of the Paper*

The paper is organized into eight sections: (a) this introduction; (b) research methodology; (c) a literature review of the evolution of the smart city concept, smart city indexes, and examples of smart cities initiatives; (d) a description of the smart city conceptual model from an infrastructure management perspective; (e) an explanation of the infrastructure management framework proposed to integrate the five dimensions of a smart city; sections about the role of technology and education; (f) a presentation of cases studies for Lima and Piura in Peru; (g) a discussion of the potential challenges of the approach, and interpretation of the goals, criteria, and variables used in the case studies from a 5D perspective, and (h) a conclusion section with a concise summary of the findings along with recommendations for future research.

## 2. Research Methodology

The main challenge is to integrate diverse knowledge and learn how to address city public demand uncertainties that are faced by local governments when seeking infrastructure solutions. To provide infrastructure solutions, considering the complexity of smart cities, it is necessary to articulate knowledge and technology from various disciplines by building knowledge management networks. Therefore, the research methodology needs to consider all these important characteristics that are intrinsic to the nature of the problem.

It is important to recall that traditional management approaches mainly focused on a single aspect of smart cities such as technology when there was a need to integrate multiple dimensions. For example, it is important to understand the social aspects of a smart city to address public infrastructure needs. As stated by Morin, "The social reality and of course of the cities, as Castoriadis maintains, is a totality that is and is not at the same time one. Today, it is crucial to reflect from the doubt, from the complexity, from the questions, and not as we are used to from the claim to provide a unique and categorical response to the problems faced by the city and its citizens. It is recognizing the difficulty; it is accepting the complexity, the uncertainty, and the need to diversify the possibilities and solutions. It is a necessary principle not only at an individual level but also at a collective level" [13].

There are two main research methods: quantitative and qualitative. In the quantitative method, data collection and analysis are used to answer the research questions expressed through a hypothesis with variables that can be measured and interpreted with descriptive and inferential statistics based on samples. In the qualitative approach, there are characters, attributes, and non-quantifiable properties; there is less emphasis on measure-

ment since questions and hypotheses arise as part of the research process, not necessarily at the beginning. The qualitative approach is based more on an inductive process, first exploring and describing the reality of the research problem to generate theoretical perspectives.

The methodology to address the research problem described in this paper follows a mixed approach and combines quantitative and qualitative research procedures due to the nature of the research question. It begins with an exploratory descriptive approach investigating the perspectives of complexity, hermeneutics, transdisciplinary, and phenomenology. Therefore, the research approach uses several techniques, including inductive–deductive methods and tools, including a comprehensive literature review, participant and non-participant observations, interviews, questionnaires, focus groups, content analysis, smart city measures, and project ranking procedures for infrastructure management. This is because of the multidimensional nature of smart cities and the complexity of management approaches to meet infrastructure demands, which require blending qualitative and quantitative methods to address the research question. The incorporation of humanistic and educational perspectives leads to the social aspects of the research problem. Research related to social problems related to smart cities traditionally adopts qualitative methods. The reason is that social problems are dynamic, changing over time as a result of a complex reality that has multiple elements with interacting facets that are constantly evolving. On the other hand, the solution to infrastructure problems involves systematic and practical methods for data collection and the development of forecasting performance models for planning and management applications. Quantitative methods of research use data collection procedures and analytical tools to approach infrastructure problems, although using these results implies assessing the impact on socio-economical aspects of alternative planning and management decisions. Hence, the mixed research approach considers the complexity of the technical and social aspects of the problem, knowing that the solutions finally implemented would affect, positively or negatively, the quality of life of all the citizens.

The authors do believe that the inclusion of humanities, sciences, technology, innovation, and education perspectives in research is essential for the formulation of questions in the context of a mixed research approach seeking solutions contextualized to the geographical and educational realities, basic needs, feelings, and emotions of the community. This reasoning resonates well with the need to foster ethical behavior among the citizens as the foundation for a better quality of life and the commonwealth of the community.

As a result, the research approach included a first phase with a comprehensive literature review and the development of an infrastructure management framework for smart cities; and a second phase with workshops and focus groups to apply the framework. To be more specific, the comprehensive literature review provides an overview of existing concepts and practices to be contrasted with participant and non-participant observations about their usefulness for infrastructure management practices. As it is explained in later sections of this manuscript, the preliminary conclusion confirmed the need for an infrastructure management framework for smart cities, pointed out the importance of a humanistic approach, and led the research methodology to the assembly of focus groups and the application of context analysis to interpret their responses. The integration of all these research techniques resulted in the development of the infrastructure management framework for smart cities. Smart city traditional measures were also reviewed, finding that they cannot be directly applied in the procedures used for infrastructure management decisions.

In the second phase of the research methodology, the Delphi method is one of the main tools proposed to collect feedback and analyze data. Figure 1 shows four stages for applying the Delphi method to the solution of smart city infrastructure problems.

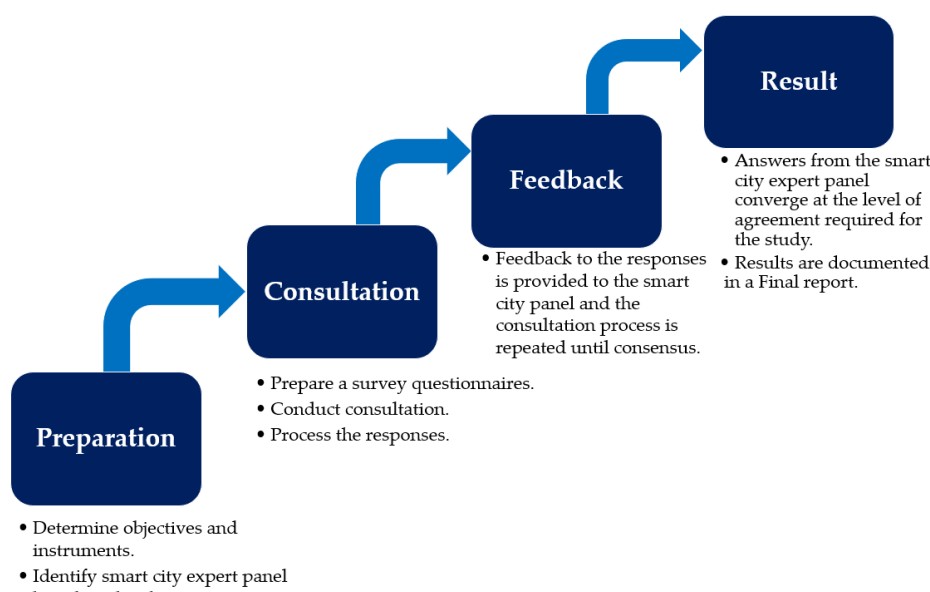

**Figure 1.** Delphi method for smart city research analysis.

A description of how to implement the four stages follows:

Preparation: Research objectives and instruments are determined to identify experts to respond to the topics related to a smart, sustainable, and resilient city. A panel is assembled, composed of experts from academia, professional practice, and citizens representing different organizations.

Consultation: A survey with questionnaires is prepared and sent to the experts to gather opinions and feedback on the identified problem. Experts can be asked to justify their answers. A follow-up workshop is conducted with practitioners who are implementing smart city initiatives in different sectors and regions, including architects, urban planners, ecologists, educators, sociologists, anthropologists, psychologists, artists linked to urban planning, sports professionals, urban heritage experts, educators, and representative citizens. The data collected in the first round are analyzed to identify patterns and trends in the expert responses and validated by the experts selected in the workshop. Statistical or qualitative techniques are used to synthesize the information.

Specific examples of questions for consultation are: How do you think about the creation of a smart, resilient, and transformative city? What technologies are relevant to help develop a smart, resilient, and sustainable city? What is the role of higher education in creating smart, sustainable, and resilient cities? How do we empower citizens so that they can understand and participate in the development of a smart city with a humanistic approach? What level of interaction is required between the city and the rural areas?

Feedback: The first results can be shared with the experts without identifying a specific identity. The experts can modify their own answers based on the information provided by their colleagues. Additional rounds are optional since the submission of feedback surveys and interviews can be done multiple times to achieve a satisfactory degree of convergence in responses until a level of consensus is achieved.

Results: When the answers converge and the arguments become consistent, conclusions are drawn.

In summary, the Delphi method collects data in terms of answers from a panel of experts whose members are identified by the research team. The panel of experts is selected in the preparation stage based on the research objectives, and questions are prepared accordingly for consultation. The results of the analysis are considered feedback to repeat the consultation process until a consensus is reached. The four stages can be put into practice with the adoption of the infrastructure management framework. The reason is that the Delphi method is a useful tool to collect qualitative data throughout the entire man-

agement process, as described in the case studies. The case studies presented in this paper describe details about the research activities, providing examples of how the methodology addressed the multi-dimensional nature of smart cities by integrating humanistic and educational perspectives that can be adapted to the particular characteristics of a specific city.

## 3. Literature Review

Smart cities arise from the convergence of two revolutions: The urbanization process and the digital revolution. In the evolution of a city to become smart, there are four phases: (1) the vertical phase of technology implementation for urban management improvement; (2) the horizontal phase of a transversal management platform connecting different services; (3) a connected phase enabling information sharing and interoperation of different technologies implemented in the vertical phase through the horizontal platform adopted in phase 2; and finally, (4) an intelligent phase offering high-value information and services to citizens and companies, creating an innovation ecosystem [14].

There is no doubt that several benefits for citizens and local businesses depend on compatible and internet-based government services that offer worldwide connectivity [15]. However, technological devices are only tools that create virtual platforms for people to use for various purposes, while the true effectiveness of their usage mainly relies on other factors such as understanding, acceptance, comfort, and their ability to contribute to preserving social sustainability. In this sense, Chen et al. mentioned four major concerns regarding social sustainability that impact the progress of a city: (a) inclusive social equity and justice; (b) quality of life with a focus on the basic needs of people; (c) level of citizen participation; and (d) human-centered smart governance [16].

As a result, the concept of a smart city has evolved to achieve livability, workability, and sustainability for all citizens. Three affirmative general approaches to the concept of a smart city are: Technocentric, socio-economic, and people oriented. The technocentric approach emphasizes technology as the driving force of urban development, while socioeconomic approaches give more weight to stakeholder participation. The people-oriented approach focuses on humanistic management principles and effective governance. On the other hand, one rejecting approach views smart cities as a technocratic dystopia followed by businesses focused on surveillance [17].

From an engineering perspective, new approaches to foster a people-centered approach in the planning, design, and management of infrastructure projects have arisen. For example, emotional engineering emphasizes the importance of meeting the expectations of the receivers or people by giving more attention to the users' emotions when planning, designing, building, and maintaining infrastructure [18]. Therefore, more recently, smart city models have enabled public engagement and social inclusion, as observed by Vienna's partnership with Wien Energy and later in the Vancouver Greenest City 2020 Action Plan with the participation of 30,000 citizens [16].

In practice, a smart city should interconnect governance, entrepreneurial initiatives, and social programs using technological tools. The objective is to boost citizens' participation in thematic networks and generate data to sustain knowledge production [19]. Interpretation of data collected in a systematic manner from integrated thematic networks should produce knowledge to formulate and implement well-informed urban plans, leading to a knowledge-driven economy by generating socio-economic value and bringing global competitive advantages. This level of development requires the development of civil infrastructure with the integration of multiple dimensions, including smart environment, smart economy, smart governance, and smart people. As a result, in order to assess the level of smartness of a city considering its multidimensional nature, smart city indexes have been developed.

### 3.1. Smart City Indexes

In 2021, The Institute for Management Development and Singapore University for Technology and Design (SUTD) ranked Singapore, Zurich, and Oslo in the first, second,

and third places, respectively, using the Smart City Index. The Smart City Index is based on economic and technology-related data and captures the perceptions of the residents about the level of smartness of their cities [20]. Examples of other smart city indexes are the IESE Cities in Motion Index (CIMI), the Innovation City Index (ICI), the European Digital City Index (EDCi), the Smart City Strategy Index (SCSI), the Global Cities Index (GCI), and the Global Livability Survey. These smart indexes mainly consider digital infrastructure and technology dimensions and assign different weights to their components or subcategories. Alderete stated that in-house experts, country analysts, and field correspondents suggested assigning a lower weight to technology as a city's digital infrastructure depends on the country's digitalization level [7].

In general, smart city indexes score different subcategories or service sectors as acceptable, tolerable, uncomfortable, undesirable, or intolerable. The subcategories are weighted to produce a smart city rating based on the location's relative performance of the city as supported by information about its service as gathered from external data sources. According to a survey conducted by the European Union in Spain, the order of priority of subcategories or service sectors as ranked by citizens living in a city is health, environment, education, security, local economy, traffic and mobility, and municipal government [14].

The International Standard Organization (ISO) has published standards to evaluate the sustainable development of smart cities. ISO is an independent and non-governmental organization, and the Technical Committees (TCs) comprise experts from over 50 countries. ISO standards for smart cities cover urban services towards the enhancement of the quality of life of the citizens, addressing various dimensions such as environment, energy, finance, education, economy, and planning [19]. ISO standards offer a framework to measure a city's smartness. For instance, ISO 37101, titled "Sustainable development in communities", outlines basic requirements for sustainable cities to prepare strategies for achieving local government goals [21]. Another example, ISO 37120 "Sustainable development of communities—Indicators for city services and quality of life", presents indicators for city services and quality of life that has two sub-categories [22]: (a) ISO 37122 describing indicators for smart cities [23]; and (b) ISO 37123 summarizing indicators for resilient cities [24,25]. Table 1 shows a list of the most common smart city indexes.

**Table 1.** Summary of smart city indexes (adapted from 7).

| Index | Number of Cities Sample | Scale | Sub-Category or Service Sector | Source |
|---|---|---|---|---|
| Cities in Motion | 181 | 1–181 (Ranking) | Governance, urban planning, public management, technology, the environment, social cohesion, transportation, human capital, and the economy | IESE School of Navarra |
| European Digital City Index | 60 | 0–60 (Ranking) | Access to capital, business environment, digital infrastructure, entrepreneurial culture, knowledge spillovers, lifestyle, market, mentoring and managerial assistance, non-digital infrastructure skills | Nesta, European Digital Forum |
| Global Cities Index | 128 | 0–100 (Ranking) | Business activity, human capital, information exchange, cultural experience, political engagement | A.T. Kearney |
| Global Livability Index | 140 | 0–100 | Stability, healthcare, culture and environment, education, and infrastructure | Economist Intelligence Unit |
| Innovation City Index | 500 | 17–60 (Score) | Cultural assets, human infrastructure (to implement innovation: transport, universities, government, technology), and networked markets (basic conditions and connections for innovation: location, military, economies of related items) | Think now |
| Smart City Index | 100 | 0–10 | Transport and mobility, sustainability, governance, innovation, economy, digitalization, living standard, expert perception | Easy Park Group |
| Smart City Strategy Index | 87 | 0–100 | Action fields (buildings, energy and environment, education, health, government, mobility), strategic planning, and IT infrastructure, | Roland Berger |

### 3.2. Examples of the Smart City Initiatives

There are many cities worldwide implementing initiatives focused on different aspects of a smart city, but they all share a common goal of enhancing the livability, workability, and sustainability of the urban environment. Examples of smart city initiatives are:

- Amsterdam: A significant approach to the city's governance is to initiate the "Smart Citizen" project involving citizens and local communities, where they work as agents of crowdsourced data with direct involvement in shaping the city as smart and resilient [26]. It represents an ideal example of smart governance, where entrepreneurs are encouraged to utilize publicly available data, design apps, and test and pilot innovative solutions to enhance services and businesses.

- Copenhagen: The city is collaborating with the Massachusetts Institute of Technology (MIT) to develop an intelligent bike system, "the Copenhagen Wheel", which is a new emblem of smart, responsive, and elegant urban mobility [27]. It ranked seventh in the IMD's list of smart cities in 2021 [20].

- New York: As a part of "Smart City Pilot Project 2020", hundreds of smart sensors have been placed in different districts, streamlining traffic flow to reduce congestion and emissions; installing clean water leakage detection systems to preserve clean drinking water; installing LED indoor farming lights; and installing advanced air quality monitoring systems. A new web-based software from HunchLab has been tested by the police department and utilizes historical crime data, terrain modeling, and other information to predict and respond to crime [20].

- Oslo: With about one million citizens, the city is focusing on creating an eco-friendly environment with smart, green transport solutions. One of the public transport companies in the city, Ruter, has declared that all its modes of transport will become emission free by 2028 [28].

- Singapore: The city is running the "Smart Mobility 2020 Initiative" towards a more connected and interactive land transport community through the development of an intelligent transportation system. Singapore's e-health initiative is driven by the Ministry of Health (MOH) and the Infocomm Media Development Authority (IMDA), which includes HealthHub, Telemedicine, TeleRehab, and robotics to efficiently provide seamless healthcare experiences for the citizens. The government of Singapore has developed a mobile app named "Smart Nation App", which creates a platform for the citizens to interact with the government and provide access to government services and data [12].

- Zurich: It has a strong human-centric policy approach with a dynamic blend of job opportunities in long-established sectors such as finance, coupled with a scene of innovation [29]. The main priorities of their smart city strategy include providing affordable housing to its residents, mitigating road congestion, solving unemployment issues, and improving air quality. In 2022, a survey conducted across 141 cities with a total of 20,000 participants gathered feedback on 15 aspects of living in their cities as well as their feelings on the adoption of smart technology, including the use of personal data and facial recognition, and what urban challenges they believe are most urgent to address. About eighty percent of the citizens rated public transport as satisfactory, while seventy percent agreed on the need for transparency to get easy access to information about local government project initiatives [29].

In the United States, Columbus, Pittsburg, and San Francisco are also implementing smart city initiatives to address transportation, sanitation, connectivity, and safety challenges in their communities. Columbus' strategies focus on better-connecting citizens to services by experimenting with innovative technologies and partnering with the private sector. The city of Pittsburgh and Allegheny County are implementing cutting-edge technologies in transportation, such as smart traffic lights that reduced aggregate waiting time at intersections by 40 percent, which helped decrease vehicle emissions by 21 percent. San Francisco's smart city strategies are focused on stakeholder engagement and encourage residents to submit proposals to the Mayor's Office of Civic Innovation [30].

## 4. Smart City Conceptual Model from an Infrastructure Management Perspective

Smart city project initiatives are usually categorized into two main approaches: "top-down" and "bottom-up". Top-down approaches emphasize technology, efficiency, master planning, and data integration from different systems into a central operations unit [31]. Conversely, bottom-up approaches adopt a more humanistic approach, focusing on citizens and their ways of using cutting-edge technologies for their benefit, such as mobile apps, social media, and open data. The bottom-up approach seeks proactive solutions to problems and promotes behavioral change and transverse critical thinking.

The authors believe that from an infrastructure management perspective, a city becomes smart when it has developed civil infrastructure systems as well as the organizational and operational capacity to fulfill the needs of its citizens. The conceptual model proposed by the authors is centered on the quality of life of citizens and is composed of five main dimensions that are integrated through an infrastructure management framework.

*5D Smart City*

In the infrastructure management approach, the authors propose five smart city dimensions (5D): Environmental, financial-economic, political governance, social-people, and technological, as shown in Figure 2. The level of development on these five dimensions determines the quality of life of the citizens.

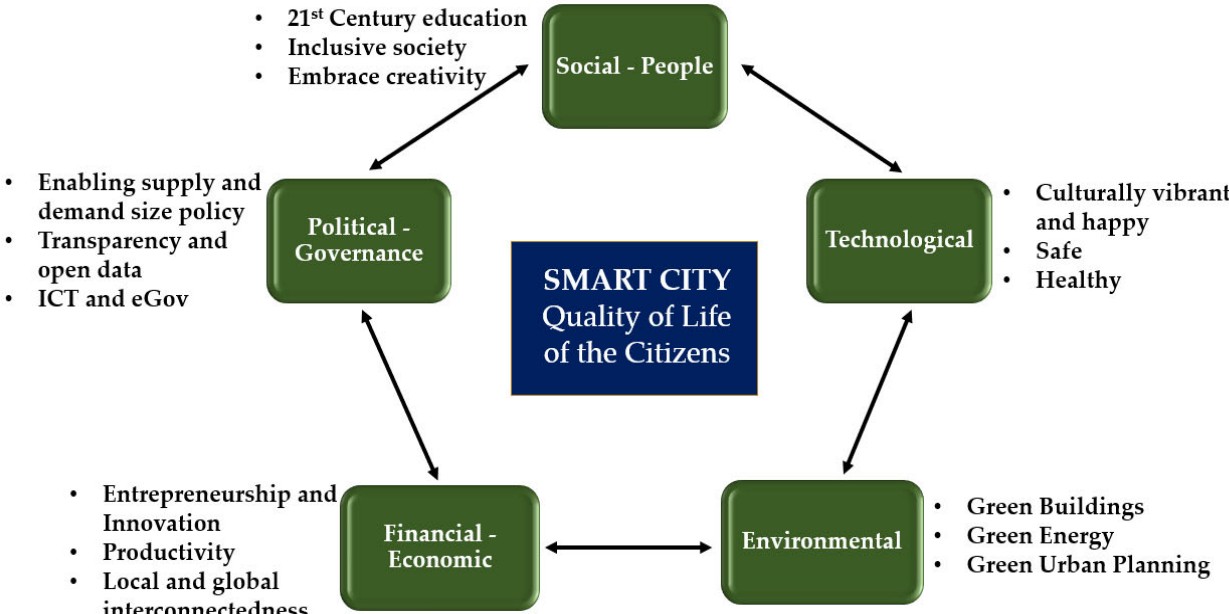

**Figure 2.** 5D smart city conceptual model (adapted from [32]).

- Environmental: Environmental concerns are getting more attention nowadays due to factors such as global warming and the rising frequency of natural disasters that pose risks for people living in urban areas. In addition, safe water supply, smart waste management, energy-efficient buildings, and green public places are required to preserve a sustainable environment for citizens.

  To be proactive and meet the evolving needs of the people while mitigating the environmental impact, innovative ecological approaches such as nature-based solutions (NbS) are adopted worldwide to solve infrastructure problems [33]. Two major challenges of the NbS approach are climate change and the impact of human activities on the planet. Different aspects of NbS include carbon storage to mitigate climate change, preservation of vegetation from rising temperatures, preventing the intensities of natural calamities, and pollution treatment.

- Financial-Economic: Local governments should work together with the private sector to create a smart economy. A smart city attracts new businesses, job opportunities, and a productive workforce, which increases productivity and workability.

  Businesses that promote resilient infrastructure, the safety of the citizens, and poverty eradication assist the city to prosper economically and accelerate the standard of living of the people. Entrepreneurs and different start-up founders take part in the development of a city's economy by making contributions to both local and global networking and infrastructure investments.

- Political-Governance: The political aspects of a smart city encompass ethical and responsible governance, which demands safeguarding data privacy and protection, ensuring cyber security, involving citizens and stakeholders in political issues affecting the management of infrastructure, and maintaining a transparent decision-making process. These elements are crucial to creating public trust in government administration.

- Social-People: The social dimensions of a smart city involve the engagement of citizens, stakeholders, leaders, and the government in the infrastructure planning and management processes; highlighting the notion that the intelligence and competence of the people are fundamental to the evolution of a smart city. Social sustainability is the central factor in creating smart functionality and flexibility of infrastructure networks. The integration of smart living, smart mobility, and smart people describes the social aspects of a smart city.

- Technological: The technological dimension of a smart city establishes an interconnected infrastructure ecosystem where individuals, from decision-makers to beneficiaries, actively engage and interact using IoT, sensors, artificial intelligence, ICT, mobile apps, geospatial technology, and blockchain. Smart city technological initiatives include energy conservation and environmental efficiencies that help reduce pollution; smart traffic management, ride-sharing services, and smart parking systems, which reduce congestion and save people's time; internet-enabled rubbish collection, bins, and fleet management systems to combat air pollution effectively; monitors and sensors providing an early warning for incidents such as floods, landslides, hurricanes, or droughts for safety measures; smart buildings offering real-time space management or structural health monitoring.

In summary, it is very important to empower citizens, along with smart applications and data analysis capabilities, with the necessary skills to effectively utilize technologies and devices in order to receive the utmost benefit from technology. For example, a comprehensive and forward-thinking virtual tool designed for a futuristic vision is Mega City 2070. It includes 39 topics, making up hundreds of callouts that can be explored by the users. The topics include artificial intelligence, hazard response and mitigation, human mobility and transit, materials, policy and planning, resilience, sustainability, water access, and management, among others. Mega City 2070 (ASCE 2022) mainly focuses on the engagement of people of all ages and across all facets of professionals from students to engineers and researchers. Therefore, it encourages stakeholders to think holistically, incorporating green buildings, green energy, green urban planning, automation, resilience, sustainability, and livability principles to foster economic and social equity [34].

## 5. Infrastructure Management Framework for a Smart City

The development, evolution, and sustainability of a smart city should be supported by an infrastructure management framework that involves setting goals and policies, building an asset inventory, assessing urban infrastructure conditions, developing and applying performance models, evaluating alternative solutions, selecting smart city project initiatives, prioritizing funding allocation, formulating short- and long-term plans, implementing the plan, and monitoring performance to tune-up infrastructure management practices.

Figure 3 shows the infrastructure management framework integrated with the five dimensions of a smart city throughout its entire process. It begins with setting goals and policies that should be aligned with the environmental, financial-economical, political-governance, social-people, and technological dimensions of a smart city. It follows the asset inventory, traditionally related to physical infrastructure facilities, but from a broader perspective, it should include human and financial resources since they are valuable assets, as emphasized in the 5D model. Condition assessment and performance modeling using forecasting methods based on statistics is the next framework component. These models are needed to evaluate alternative solutions that must assess the five dimensions to formulate short and long-term plans considering the available budget allocation and financial capacity. A successful program implementation phase is unfeasible without the integration of the five dimensions into the projects. Once the program is implemented, monitoring of the results, in terms of performance, is required. Monitoring the performance of smart cities requires following the responses using measurements that are able to proactively capture changes in the five dimensions in order to tune-up infrastructure management practices. Therefore, lessons learned from the program's implementation should provide feedback to potentially modify goals and policies. During the entire infrastructure management process, the participation and interaction of local authorities and the public through effective communication tools are essential to ensuring a balance among the five dimensions.

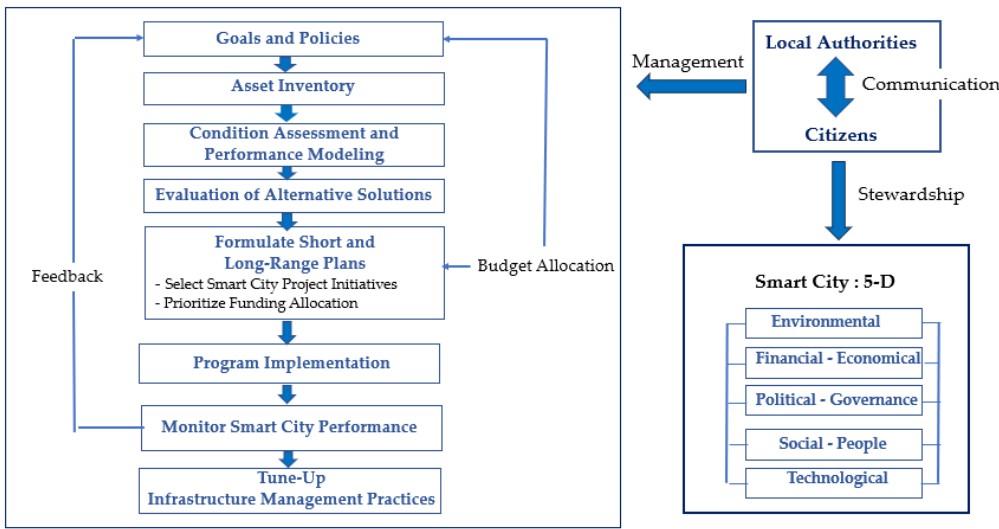

**Figure 3.** Overview of an infrastructure management framework for smart cities (adapted from [35]).

The implementation of a holistic interactive approach with the participation of local authorities and the public in the infrastructure management process will contribute to sustaining and strengthening the smart city 5D model. Infrastructure management strategies supported by specific standards and practical guidelines should enhance the overall project selection process, prioritizing the resources allocated to smart city initiatives. There are different approaches for project selection and decision-making prioritization analysis. Wua and Chen used multi-criteria decision making (MDM) to identify alternative solutions and establish evaluation criteria for project selection using the analytic hierarchy process (AHP) to assist decision-makers in assigning weights to the dimensions taken into consideration in the selection criteria [36].

Project prioritization for resource and budget allocation methods should not only involve local government authorities but also the citizens. Different approaches to gathering feedback, such as the Delphi method, can be adapted to allow citizen and expert participation in the management process. For example, Preble mentioned that "Delphi can act as a useful decision and that can supplement the more traditional planning and budgeting methods and models", examining its application to address problems related to environ-



mental concerns; urban transportation; urban fiscal policy or municipal finance; and food production, marketing, and distribution in the cities [37].

This interactive management approach involves iterative stages, introducing questionnaires to ensure comprehensive feedback from the participants, who play an important role in project prioritization [38]. Moreover, the participation and collaboration of local government, private companies, universities, and most importantly, citizens are required as the main component of the methodology to construct a smart city ecosystem in an urban environment. In this line, Sekayi and Kennedy recommended a Likert-type questionnaire response to facilitate the feedback and analysis of the responses of the participants in a Delphi study [39]. This questionnaire complements the open-ended questions by gathering qualitative feedback from the participants using a five-point Likert-type scale that can be transformed for quantitative analysis (e.g., 1: Strongly disagree, 2: Disagree, 3: Neither agree nor disagree, 4: Agree, 5: Strongly agree), repeating the consultation and feedback process until consensus is reached. Regarding the analysis of the responses, Keeny states that "there is no standard approach used to analyze data from Delphi rounds" [40]. Therefore, the researchers propose descriptive statistics to analyze the answers to Likert-type questions and content analysis to identify frequent words in the responses to open questions.

It is hoped that the combination of traditional tools used in infrastructure management practices with novel approaches will result in an enhanced planning and management process for the benefit of the citizens. One aspect that needs to be carefully considered is the performance measures to be used to support the project selection process for funding allocation. This opens research that merits being studied by itself, although this paper provides initial brainstorming and discussion to ignite further development. For this reason, the importance of a sustainability rating system and the roles of technology and education to support the infrastructure management framework towards the development, evolution, and sustainability of smart cities are emphasized in the following sub-sections.

### 5.1. Sustainability Rating System: Envision^TM

Infrastructure planning and management should be designed to be adaptable and flexible to the world's changing conditions, such as climate change, natural disasters, and technological advancements. A smart city should uphold resilient civil infrastructure to withstand the shocks and stresses created by disasters, and cyber security threats, and promote sustainable urban development.

Envision is a ranking tool to be considered as an alternative for the selection of sustainable projects. Envision is a comprehensive ranking tool that enables a thorough examination of the sustainability and resiliency of all types of civil infrastructure [41]. Envision leads to a number of benefits by promoting environmental justice and social equity, aiding carbon neutrality, enhancing stakeholder engagement, fostering infrastructure resiliency, enabling climate-ready infrastructure, and seeking cost-effective and resource-efficient projects. More than 200 cities, counties, public agencies, academic institutions, 250 private companies, and industry associations have used Envision.

Envision comprises a framework of five categories: Quality of Life (QL), Leadership (LD), Resource Allocation (RA), Natural World (NW), and Climate and Resilience (CR) that consists of 64 sustainability and resilience indicators or "credits" under different subcategories. For instance, there are three subcategories and fourteen credit indicators under QL, including Improve Community Quality of Life, Minimize Construction Impacts, Improve Community Mobility and Access, Advance Equity, and Social Justice, among others [42]. Table 2 shows the five Envision^TM categories along with a total of 14 subcategories and their corresponding maximum points [43].

**Table 2.** Envision$^{TM}$ categories, subcategories, and points table [42].

| Category | Subcategory | Maximum Points | |
|---|---|---|---|
| Quality of Life (QL) | Wellbeing | 92 | 200 |
| | Mobility | 44 | |
| | Community | 64 | |
| Leadership (LD) | Collaboration | 72 | 182 |
| | Economy | 60 | |
| | Planning | 50 | |
| Resource Allocation (RA) | Materials | 66 | 196 |
| | Energy | 76 | |
| | Water | 54 | |
| Natural World (NW) | Sitting | 82 | 232 |
| | Conservation | 78 | |
| | Ecology | 72 | |
| Climate and Resilience | Emissions | 64 | 190 |
| | Resilience | 126 | |
| Total points | | 1000 | |

Stakeholders can evaluate projects through the credit indicators of Envision, which consider five different levels of achievement in the criteria: Improved (performance above conventional); Enhanced (performance adheres to Envision$^{TM}$); Superior (high-level performance); Conserving (Performance with zero impact); and Restorative (a performance that restores systems).

The Envision categories fit into the 5D conceptual model described in this paper. The natural world and climate and resilience Envision categories are considered in the environmental dimension of the 5D model; the resource allocation category is aligned with the financial-economical dimension of the 5D model, but it could also be related to the political-governance dimension; the leadership category is related to the political-governance dimension; and the quality of life category is related to the social-people dimension. The technological dimension of the 5D model is not explicitly defined as an Envision category, although it is required to assess the feasibility of implementing a smart city initiative. Further comments and ideas on how to handle the selection or development of smart city indicators for infrastructure management practices are provided in the discussion section of this paper.

### 5.2. Role of Technology in Smart Cities

The progress of smart cities is greatly influenced by the advancement of technologies such as blockchain, building information modeling—city information modeling (BIM-CIM), geospatial technology, smart Internet of Things (IoT) devices, and twin digital models. A brief description of these technologies follows.

- Building Information Modeling-City Information Modeling (BIM-CIM): Building information modeling (BIM) is a 3D modeling tool for urban planning and design that helps a city improve operational efficiency, assist stakeholders in decision making, and mitigate risks and vulnerabilities. The tool enables collaboration among architects, engineers, planners, and stakeholders by providing a platform to share data and information, thus creating interconnectivity among parties involved in a specific project. On the other hand, a geographic information system (GIS) is a platform to store urban data and information about the locations of buildings, topography, and occupancy. The development of city information modeling (CIM) is a relatively recent idea that was proposed to enable multi-hazard simulation using a unified database covering both individual buildings and urban areas. City information modeling (CIM) combines the spatial data representation of a geographic information system (GIS) with the richness of expressing individual building component information in BIM. Figure 4 shows the concept of city information modeling (CIM) integrating BIM and GIS [44].

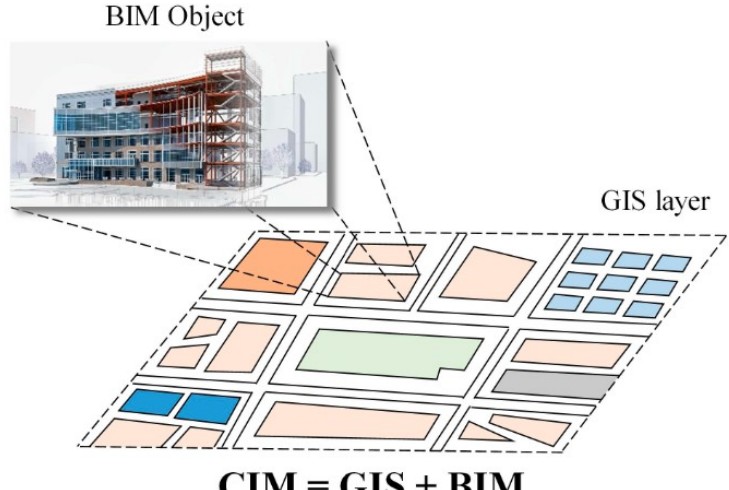

**Figure 4.** CIM integration of BIM and GIS [36].

- Digital Twin Model: Digital twin is a virtual simulation model that replicates physical features, and objects, and captures the process in real time, providing a platform to compare the planning or design of project initiatives with the current real-time situation. An example is the digital twin of Maracaibo, Venezuela's second-largest city, with a bird's eye view down to specific buildings. It was developed by ArcGIS Urban and related solutions such as CityEngine, Esri Venezuela, and partners at the University of Zulia. This digital twin incorporates indicators and variables such as power consumption, mobility patterns, environment, and zoning regulations, allowing assessment of different scenarios and evaluation of plan alignment with public policy goals [45].

- Geospatial Technology: Geospatial technology is a multidisciplinary area that includes various technologies such as GIS, global positioning systems (GPS), and remote sensing. Maps created using geospatial technology assist decision-makers in visualizing and identifying problems. While performing advanced geospatial analysis, it is necessary to maintain and preserve large-scale data for intergovernmental coordination related to smart cities. Many smart city initiatives are using geospatial technology to identify areas where existing infrastructure systems are inadequate. For instance, the San Diego Association of Governments (SANDAG), California, is using GIS for regional planning and transportation projects to assess the existing conditions and identify stresses on the transportation network in order to learn the current and future needs of citizens and provide transportation alternatives that promote equity [46].

- Smart (IoT) devices: Using different devices such as smart sensors, monitoring devices, visibility devices, and AI for receiving and managing big data efficiently, IoT is enabling seamless urban interconnectivity and communication between different systems and infrastructures. IoT applications have made ground-breaking changes in different areas such as traffic control systems, energy consumption, and waste management. These advancements have resulted in optimizing resource allocation processes, reducing pollution, and saving time as a result of the adoption of sustainable practices. Cloud-based (IoT) applications facilitate real-time data management for citizens using smartphones in different areas of urban eco-systems with the participation of municipalities and enterprises. The Oslo Smart Street Lighting project is a prime example of an IoT application in smart cities. By incorporating smart sensors and utilizing internet-based apps, Oslo integrated the city's street lighting into a remotely accessible network, resulting in approximately 70% energy savings through the deployment of 20,000 smart streetlights [47]. Figure 5 shows the various roles of IoT in smart cities by using different devices.

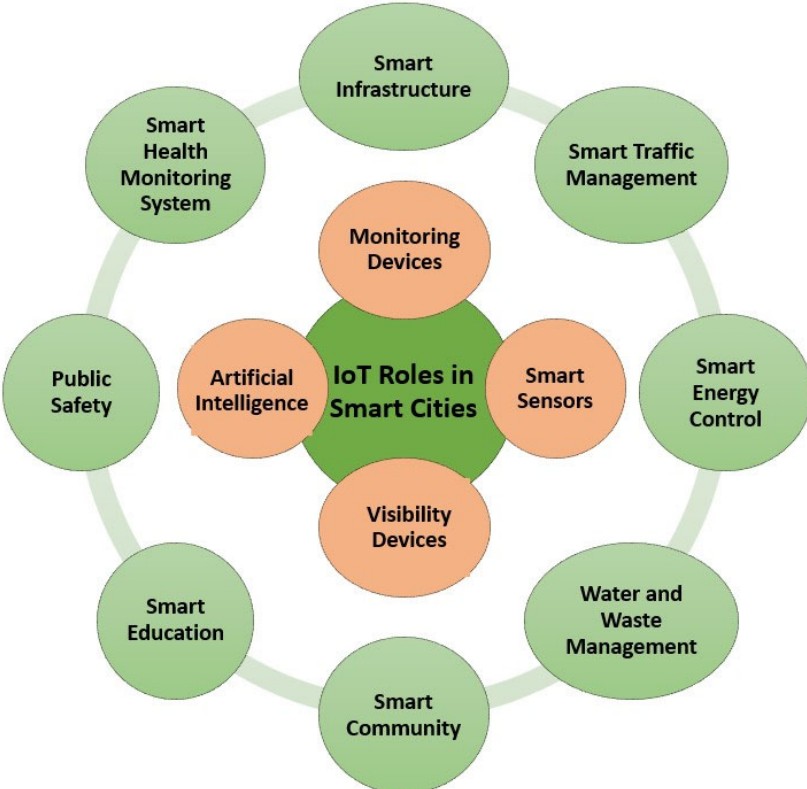

**Figure 5.** IoT roles towards smart cities.

Despite the benefits brought by smart (IoT) devices, new economic and social opportunities arose, posing two entangled main challenges in the area of security: (a) illegal access to information and (b) privacy digital intrusion when citizens share their location and activities [48]. With the increasing use of interconnected smart IoT devices, smart cities need to have protection against cybersecurity risks such as device hijacking, data theft, man-in-the-middle attacks, distributed denial of service (DDoS), and permanent denial of service (PDoS) [49]. It is important to follow proper algorithms, frameworks, models, and protocols to improve data privacy and cybersecurity when launching any new technological advancement. For instance, a cloud-oriented testing architecture can be incorporated to execute tests and collect testing results for IoT in smart cities [50].

### 5.3. Role of Education in Smart Cities

The most important component of a smart city is its citizens, who should benefit from the progressive enhancement of the services provided by the city. Local governments can initiate many projects, but any project will fail if the citizens do not accept and collaborate with it. A smart city requires a local government with city planners, managers, and citizens working together with the support of a common technological platform to develop, select, and implement project initiatives to preserve safe, secure, and healthy urban infrastructure systems.

Citizens are required to participate with ethics, responsibility, and solidarity in the development and implementation of smart city projects. For this reason, education is essential to forming smart citizens. There are many definitions of education, Pablo Natorp stated that "There is only one educational entity: the community. An individual educational action is as impossible as the existence of a man dispensing with society" [51]. In addition, Hans Thiersch affirmed that: "The role of the social pedagogue is to help people to critically analyze their problems, reflecting on the social causes of individual problems and to find options for a successful everyday life. The focus is connecting help for the individual with political action in the context of social justice and well-being while recognizing social and political resources" [52,53]. The infrastructure management humanistic

approach for smart cities emphasizes the importance of social interaction for education as a foundation for the sustainability of life in all its manifestations.

Education is a process inherent in the human condition. It is a way of understanding the world that is valid in a community and determines all the work of being human. It occurs in a network of conversations and interactions that coordinate the knowledge, the doing, and the emotions of the participants and has as its supreme purpose to forge integral citizens, with a humanistic vision of society, committed to the environment.

"Quality Education" is goal 4 of the SDGs stated by the United Nations [4]. Education, among other aspects, must make human beings aware that: "The essence of the human condition cannot be dissociated. We are at the same time individuals, society, and species, you cannot have individual health if there is no public or community health, and that will be only achieved to the extent that humanity understands the fabric of life in nature and the fabric of life in political, social systems, cities, health, etc., without which there is no sustainability. On ethics and love for humanity and nature, with which we interact in the world, our survival on planet Earth will depend" [54].

Citizens are valuable assets, and their education plays a crucial role in developing a smart city since conventional education has limitations. Education based on positivist and reductionist thinking, reinforced by instrumental reason and the commodification of life, can well be held responsible for the predatory action of humans on nature and their blindness to understanding and addressing the serious problems of humanity [55]. From this perception, knowledge was dehumanized, imposing a single way of creating knowledge and making invisible other cultures and ways of being in the world.

Further development of the sciences made it clear that some of the emergent properties of life and nature could not necessarily be addressed by deterministic and reversible science. Such understandings, which have been evolving, have produced changes in the ways of thinking and living in the world, the structures of knowledge, and the way of conducting science. They foster the rise of complex sciences and complex thinking, which have integrated real-world irreversibility into science [13]. A reform of thought is required; the fragmented and deterministic dominant thinking that isolates and separates must be replaced by the complex thinking that unites and distinguishes. Morin proposes a new epistemic framework to think, feel, and act in the world [55].

From this epistemic framework, it is observed that most cities are managed from a linear and fragmented perspective, made only for commerce and transport business purposes. It is necessary to think about networked cities where different agents and multiple variables interact at the same time while providing dynamic feedback as the city evolves. For Morin "cities need to promote an inclusive, equitable and solidary urban development in which the articulation between the individual, the species and society is the vision and mission of a generous and responsible governance. Since ancient times, the city is not a problem as such; on the contrary, the urban environment is a human creation substantive for peaceful coexistence, the emergence of commerce, politics, institutions, law, civilization, the arts…" [55].

At present, the adoption and management of technological tools and social networks have affected the social life of the citizens. Maturana (2014) stated that to educate is to create, perform, and validate a particular form of coexistence [56]. This is always conducted in a network of talks that coordinate the thoughts and emotions of the participants. This means giving every citizen the elements for an autonomous, socially, and ecologically responsible task. In today's age of global connectivity and virtual networking, people from every profession now seek flexibility in work and networking from any place in the world. Moreover, in order to achieve the goals aimed at improving the quality of life of the people, a skilled and educated group of citizens, who actively embrace social innovation in industry is required.

In this sense, collaborative education is setting a new era for an education environment where students get access to open data, ideas, and work from other societies, receive suggestions from different experts in real-life scenarios, and build knowledge using IoT

and other technologies that are expanding day by day. Smart education creates a path of innovative and creative culture, enabling people to generate new ideas through lifelong learning, which flourishes the economic growth of a country. Such an evolving education system has accelerated changes in the structure of society and the thinking processes of its citizens. Social interactions and conversations create a network of knowledge, capability, tasks, and living emotions in the participants, resulting in the education of citizens with a humanistic vision of society.

Hence, it is a fundamental condition to form an informed, critical citizenry, with digital, communication, and social interaction skills. Schools must be transformed into advocates of culture and reading, reinforce the teaching of humanities, sciences, technology, and innovation in formal and non-formal settings, promote public spaces for art and sports, and enhance the aesthetics of the urban landscape, where the use of the infrastructure for living healthy is achieved.

Finally, it is worth emphasizing that ethical behavior is the foundation of education. Ethical principles are crucial to forming responsible citizens, aligning them with moral codes of practice to respect cultural values while promoting equitable living standards to develop and sustain inclusive smart cities.

## 6. Case Study: Project to Develop Smart, Sustainable, and Resilient Cities in Peru

Peru is currently undergoing many smart city initiatives, and two case studies for Lima and Piura are described in this section. The two case studies complement each other to illustrate the research methodology for the implementation of the infrastructure management framework as described in this paper for smart cities. The first case study, city of Lima, focuses on explaining how the five smart city dimensions can be used to identify challenges and propose activities to address them. The second case, city of Piura, provides more details on the procedures and products as a result of the implementation of the methodology.

### 6.1. Approaches for a Smart and Sustainable Urban Future for Lima, Peru

Lima has many socioeconomic, environmental, and cultural drawbacks, and the National Institute of Statistics and Informatics (INEI) indicates that most of the Lima's population expresses feelings of unsafety (89%), disapproval of public transportation (60%), dissatisfaction about healthcare services (86%), lack of formal employment (70%), feelings of insecurity during an earthquake (70%), and dissatisfaction with the quality of public services (57%) [57]. In response to this situation, PLANMET 204 has been prepared.

### 6.1.1. PLANMET 2040: Lima, Peru

PLANMET 2040 was developed by the Municipality of Metropolitan Lima, the Metropolitan Institute of Planning, and the Ministry of Housing, Construction and Sanitation, which presented Specific Agreement No. 606-2020-VIVIENDA in August 2022 [58]. This plan consists of implementation and management instruments based on a comprehensive urban characterization of Lima that enables the visualization of the current urban dynamics and trends within the city, considering both internal and external territorial factors. The purpose of the plan is to improve the quality of life of the citizens, preserve the natural and cultural heritage, identify, and reduce the conditions of risk areas in the territory, and prioritize investments across the three levels of government: national, regional, and district.

### 6.1.2. Expanding the Smart City Vision for Lima

To address the multidimensional problems of Lima, an epistemic and humanistic approach should be adopted to understand and manage the complex dynamics of the city, where the interacting agents are constantly adapting and co-evolving. This indicates the necessity of the transformation from a conventional to a smart education system, with the implication of new technologies for managing and interacting between different sub-

systems such as education, culture, transportation, energy, security, and supply chains. The development of Lima as a smart city requires multidisciplinary, interdisciplinary, and transdisciplinary approaches. A lifelong education with an understanding of the human condition is a prerequisite to preserving nature and culture. Learning how to live together with solidarity, civic identity, and a well-being community orientation.

Three main initiatives are proposed for Lima to become a smart, sustainable, and resilient city:

1.  Development and implementation of an infrastructure management humanistic-centered approach prioritizing livability, planning, design, and management among citizens.
2.  Building a technological digital platform that enhances infrastructure management efficiency.
3.  Integrating environmental principles into practical local government regulations that support sustainable infrastructure management.

To further develop these initiatives, the 5-dimensional aspects of a smart city should be adapted to the local context to establish a practical infrastructure management model with specific activities for Lima.

### 6.1.3. Delphi Method

It is very important to know what citizens think about health, education, the economy, and overall quality of life; to know about smart city goals and technologies; and to know their expectations for the living conditions and services provided by urban civil infrastructure systems.

The Delphi method was adopted for the smart city research project in Lima. This is a well-established collaborative research method that allows for the reflection of the participants as they respond to inquiries posed by research experts [35]. In order to apply the Delphi method to the smart city project in Lima, an expert panel should be assembled to bring multiple perspectives on the areas of transportation, energy, housing, citizen engagement, and sustainability, to name a few. Participants should be composed of experts but also citizens who can provide opinions or judgments through surveys or questionnaires. Table 3 describes examples of challenges already identified for the city of Lima.

Subsequently, the responses should be analyzed to prepare a report that can be distributed to urban planners, local authorities, and stakeholders who are involved in the project initiatives. Feedback from the responses can be provided to the participants, and the consultation stage can be repeated until a consensus is reached. Workshops, capacity-building activities, and environmental education activities should be organized to foster participation and awareness of the initiatives among citizens.

### 6.2. Sectorial Planning Approaches for a Sustainable Urban Infrastructure Future in Piura, Peru

The city of Piura, located in Northwestern Peru, is a central hub within the region, where the primary income source comes from agriculture and fishing. The city of Piura has been experiencing rapid urban growth, leading to a migration of people from rural to urban areas since the 1970s. The urban expansion resulted in the loss of agricultural land, a reduction in biodiversity, increased energy consumption, and air pollution from heavy traffic. Moreover, due to its geographic location, particularly its proximity to the riverbanks, and the presence of a blind water basin, the city center of Piura has become highly susceptible to natural calamities such as heavy flooding.

Following a multidisciplinary survey and analyses aligned to the 5D model, Morgenstadt City Lab methodology developed strategies and roadmaps for sustainable urban development in the city of Piura [59]. The analyses and proposed solutions incorporating resilience and ecological goals alongside prospects for economic and social innovation are described in the following sections.

Table 3. Examples of proposed solutions for addressing Lima challenges from a 5D perspective.

| Smart City Dimension | Challenge | Proposed Activities |
|---|---|---|
| Environmental | Deforestation | • Identifying a suitable location to implement a pilot project of camping, including the citizens for programs such as planting. |
| Financial-economical | Poverty eradication and increasing productivity | • Initiating an interconnected approach where both government and private sectors will work together.<br>• Entrepreneurs should be inspired to generate new ideas for businesses and initiatives of smart city. |
| Political-governance | Transparent decision-making process | • Transforming conventional government to a digital smart government by initiating IoT, big data, etc.<br>• Initiating open information system where citizens can see every initiative the government. |
| Social-people | Implementing Smart Education | • Explore a conceptual model from the perspective of complex systems<br>• Collaborative education system focusing on healthcare, transportation, citizen security, energy, building management.<br>• Investigating a proposal for civic education from a complexity approach |
| Technological | Adaptation of modern technology | • Proposing the application of building information modeling (BIM) as a tool to improve collaboration and communication among stakeholders involved in urban infrastructure design and management.<br>• Proposing workshops for inspiring people for adapting technological use and education. |

### 6.2.1. City Lab Methodology

Cities worldwide need to manage and adapt to global climatic change and technological shifts. They are facing challenges in breaking up past conventional practices while embracing the challenges. The City Lab of the Morgenstadt Global Smart Cities Initiative (MGI), an approach developed by the Fraunhofer Society, seeks a comprehensive and systematic understanding of how a city is functioning by implementing Morgentadt tools using quantitative and qualitative methods that involve close collaboration with its stakeholders, ranging from city council members and department heads to private companies, research professors, and local government representatives [60]. An integrated roadmap to achieve a smarter and more sustainable city is then proposed based on the findings of the analyses. The framework of City Lab is structured around an ecosystem approach that resembles the infrastructure management framework, although it is focused on three specific levels of analysis: governance, technologies and infrastructure, and socio-economic strategy. The process of City Labs has proven that achieving "smart" urbanism involves a holistic understanding of the complexity of urban systems in the implementation of high-tech solutions.

Using performance indicators, a digital on-site assessment was conducted with key stakeholders in Piura to gather data to identify the main challenges, opportunities, and influential factors that impact the development of initiatives, projects, and programs de-

ployed in the city. Data were gathered and analyzed using different methods such as interviews, workshops, surveys, and content analysis, which are part of the Delphi methodological research approach.

The assessment of the data is a qualitative analysis of the essential fields of action identified for Piura's sustainable development. It shows the actions taken by the city to address its own sustainability challenges. The action fields model analyzes seventeen essential aspects of the urban environment, including education, ICT/data governance, transport/mobility, regulations and incentives, municipal planning and strategy, buildings, urban planning, urban regeneration, resilience engineering, business tactics, water, green and blue infrastructure, organization and structure, finance and procurements, solid waste and resources, energy, and research and development tactics. Each of them is assessed through three to five questions, which, in sum, are worth 10 points when answered affirmatively. Therefore, if all questions are answered affirmatively, an action field receives 10 out of 10 points. However, if some questions are answered with a "no", the total score of the field of action is reduced. The result of the action field profile serves as the basis for developing a roadmap. The action fields are critical to inform the roadmap development process as well as provide useful information to local experts, stakeholders, and policymakers [61].

The analysis of the most critical fields of action are categorized into three clusters that involve three major actions that are foreseen as critical to the sustainable urban development and climate change resilience response of the city. These clusters are: (a) establishing digital connections and tools to improve resource efficiency; (b) the utilization of unique energy opportunities for a carbon-neutral city; and (c) the need to work on how to plan and implement sustainable development initiatives already identified for the city of Piura.

A cross-integration of the city system analysis and sensitivity analysis, together with discussions and workshops, has generated a total of 35 project ideas. From these 35 ideas, a selection of 12 was made, as it is believed they can have a strong impact on the sustainable development of Piura. The list of projects shows a mix of project ideas based on existing initiatives in Piura as well as the information received during the on-site assessment interviews that incorporate features and proposals not discussed in Piura before.

Each of the proposed project ideas was evaluated using a project ranking tool based on the 11 criteria. The criteria are:

- Alignment with the city's objectives
- Stakeholder engagement
- Replicability potential
- Regulatory constraints
- GHG mitigation potential
- Climate change adaptation potential
- Need for financial support from the public sector
- Likelihood of obtaining public funding in support of the project
- Interest in the participation of private sector financial support
- Project approval risk
- The extent of associated resettlement and rehabilitation issues

The objective of the analysis was to examine which project ideas have the greatest potential to meet Piura's needs. Each of the criteria is assigned a weighting, which indicates the relevance of those criteria: high, medium, and low.

6.2.2. Sustainable Urban Measures for Piura

The City Lab provided an action-oriented roadmap after conducting a sustainable city profile and a detailed analysis of specific urban sectors in Piura. The roadmap serves as a guideline for executing innovative measures and projects that address greenhouse gas (GHG) emissions, climate change adaptation, and the preservation of biological diversity, in the context of smart city concepts [59].

Figure 6 shows the strategic roadmap for the sustainable development of the city of Piura. It shows the organization of the project ideas into three distinct categories based on a timeframe for implementation: physical interventions, represented by oval-shaped boxes; planning projects, represented by rectangular boxes; and currently developing city projects, represented by diamond-shaped boxes. 12 projects from the first category have been selected for urban development within the city of Piura. The proposed measures integrate ecological and resilience goals that are based on the concepts of smart and resilient cities. They are related to three sectors: energy, water, and urban planning, since they are seen as the most important for urban development in Piura, with governance as an overarching field [59].

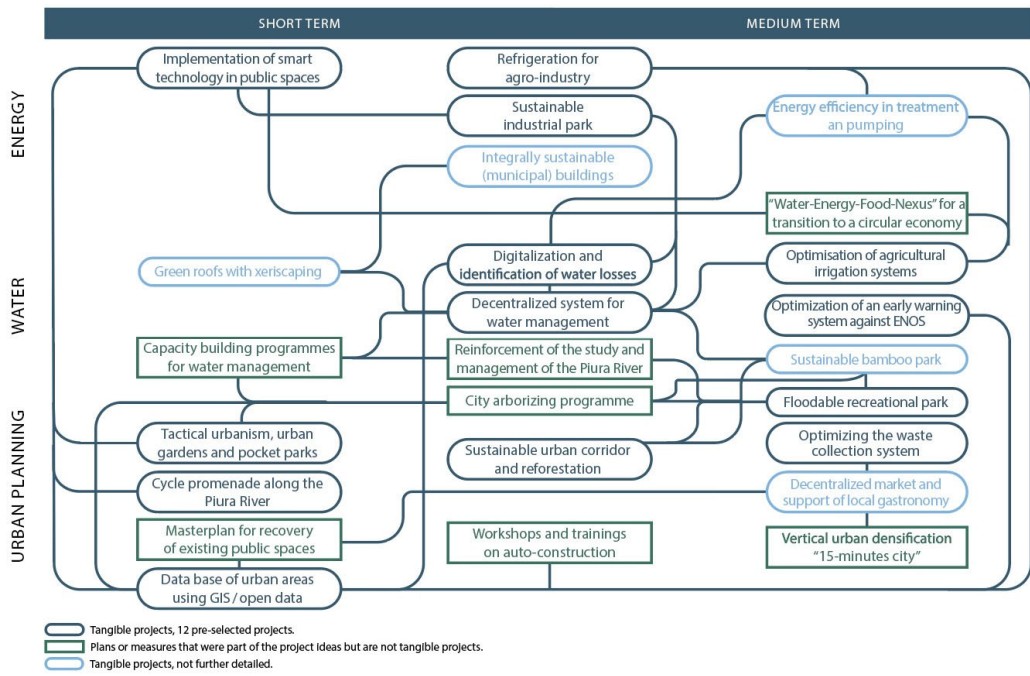

**Figure 6.** Strategic roadmap for sustainable development of Piura [61].

Some of the initiatives of the Morgenstadt Global Smart Cities Initiative (MGI) project include the transformation of public places in Piura to sustainable recreational parks, more urban gardens and pocket parks, the implementation of decentralized wastewater treatment systems, open data, the optimization of agricultural irrigation systems, offering job opportunities to the marginalized population, the implementation of smart technology in public spaces, and refrigeration for the agro-industry [59].

These project initiatives will strengthen the evolution of a smart and sustainable city over time by implementing strategies that consider a circular economy approach. In addition, a collaborative or mutual learning system should be established using education and training tools. As part of the MGI initiative, several participatory workshops, capacity building and environmental education programs, and some festivals have been organized for Piura's citizens, aimed at increasing their participation to build a sense of responsibility while raising awareness of the practical benefits of the program for the commonwealth of the community living in the city [62].

## 7. Discussion

For a people-centered design of a smart city, it is very important to incorporate an infrastructure humanistic approach that integrates the perception of citizens in the planning and management of a city. Surveys and questionnaire methods using both Likert-type and open-ended questions can help in this regard. Examples of questions that can be asked about a smart city are:

- What are the major expectations of the citizens from the civil infrastructure systems in their city?
- How satisfied are the citizens with the services provided by the civil infrastructure systems?
- What are the main city aspects that should be prioritized when developing infrastructure projects?
- Should civil infrastructure project initiatives ensure public safety and security?
- Are citizens comfortable using the new IT tools? Why or why not?

Survey responses can be complemented with smart city indexes. For example, two of the most trending smart city indexes are the IMD-SUTD Smart City Index (SCI) and the IESE Cities in Motion Index (CIMI). These two indexes evaluate the smartness of a city based on the environment, sustainability, economy, governance, social cohesion, human capital, public management, living standard, expert perception, and technology, which cover the five dimensions of the humanistic model. There is also the Envision sustainability rating system available, as described in a previous section of this paper.

Despite the many methods and indexes available for smart cities, Chai Keong Toh stated in a study published in 2023 that: "Currently, there are not uniformly and universally accepted methods for comprehensive and fair evaluation of smart cities. This is a problem as no ranking is widely accepted and universally agreed upon" [63]. This was the conclusion of an in-depth investigation and analysis of smart city indexes, criteria, indicators, and rankings. Hence, regarding the adoption of a smart city index for infrastructure management, it is noted first that the criteria used by the existing smart city indexes can be clustered using the 5D model, although the shortcoming is that none of these indexes can be used directly for infrastructure management applications. For example, the Cities in Motion index uses nine criteria with 101 indicators: Human capital, social cohesion, economy, urban planning, environment, international protection, and technology. Human and social cohesion criteria correspond to the social dimension in the 5D model, economy criterion to the financial and economic dimension, urban planning and international protection criteria overlap with all five dimensions but in particular with the political-governance dimension, environment criterion with the environmental dimension, and technology criterion with the technological dimension. Another example is the smart city index, which has five criteria: Activities, opportunities, health and safety, governance, and mobility. Here, activities refer to the easiness of citizens to enjoy entertainment and relaxation, whereas opportunities refer to schooling and work opportunities. The activities criterion corresponds to the social dimension in the 5D model; the opportunities criterion to the financial-economical dimension; health and safety criteria overlap with environmental and technological; the governance criterion with the political-governance dimension; and the mobility criterion with the financial-economical, social, and technological dimensions.

There are smart city indexes already developed with criteria that fit into the 5D model; however, infrastructure management requires condition assessment metrics based on field inspections. For example, the Pavement Condition Index (PCI) is used by local governments to manage pavements for street networks. PCI ranges from 100 (excellent) to 0 (very poor), and it requires collecting surface distress data that are not considered in any of the smart city indexes. The Metropolitan Transportation Commission (MTC) of California has developed Condition Index Distress Identification Manuals for Flexible and Rigid Pavements [64,65].

Therefore, the shortcomings of the existing smart city indexes depend on the difficulty of translating their criteria into a practical index for infrastructure management practices (e.g., percentage of urban population with adequate sanitation services, renewable water resources, number of photos of the city uploaded online) and assessing their usefulness in the planning and management decision-making process of civil infrastructure. In this sense, the main challenge is to review and condense these criteria to merge them with practical indicators used for infrastructure management.

It is worth mentioning that research studies have been conducted in the past to incorporate livability principles expressed in terms of connectivity, quality, proximity, and

safety into a transportation management framework. For example, research conducted by Chang proposed a weighted effectiveness ratio that is calculated with Equation (1) [66]:

$$\text{WERLIV} = 1000 \times \text{IMP\_AS} \times \text{IMP\_LOC} \times 1/(\text{RL\_AT}) \times (1/\text{EUAC}) \qquad (1)$$

where:

IMPAS = asset importance index;
IMPLOC = location importance index;
RLAT = remaining life after treatment or construction;
EUAC = equivalent uniform annual cost, calculated as

$$\text{UAC} = \text{COST\_F} \times (f\,[\![(1+f)]\!]^{\,n})/([\![(1+f)]\!]^{\,n-1}) \qquad (2)$$

where COST_F = COST_P $\times ((100 + f)/100)^n$

n = years of analysis, equals to RLAT or number of years from first analysis year to year of treatment;
f = inflation rate (in %)
COSTF = future inflated costs (unit costs at analysis date);
COSTP = present costs (unit costs current at the first analysis year).

Projects are ranked from highest to lowest WERLIV, and the available budget is allocated using the dynamic bubble-up technique DBU [67].

It is also possible to use selected criteria from existing smart city indexes to compose a Smart City Infrastructure Priority Index (Smart-ICIPI). An example of how to incorporate criteria from existing indexes is the development of the safety-weighted effectiveness ratio (SWER) proposed by Chang to prioritize funding allocation in asset management practices while integrating safety criteria for vulnerable road users (VRUs) [68]. VRUs are pedestrians, bicyclists, and motorcyclists, and the SWER equation.

$$\text{SWER} = 1000 \times \text{API} \times \text{ALI} \times 1/\text{EUAC} \times 1/(\text{RL\_AT}) \times \text{VRUSI} \qquad (3)$$

where:

API = asset priority index;
ALI = asset location index;
EUAC = equivalent uniform annual cost;
RLAT = remaining service life after treatment;
VRUSI = vulnerable road user safety index.

VRUSI is a metric to assess the safety conditions of road infrastructure for vulnerable road users. VRUSI combines three existing specific indexes: (a) the pedestrian level of comfort (PLOC), (b) the pedestrian level of traffic stress (PLTS), and (c) the pedestrian intersection safety index (Ped ISI).

A similar approach can be used for the development of the Smart Infrastructure Condition Investment Priority Index (Smart-ICIPI) using the 5D model combined with existing smart city and infrastructure management indexes or criteria selected from them. The equation for this index should be a function of parameters representing the five dimensions of smart cities and infrastructure conditions (see Equation (4)).

$$\text{SmartICIPI} = \text{Function } (E, W_E, F, W_F, P, W_P, W, W_S, T, W_T, \text{ICI}, W_{IC}) \qquad (4)$$

E: environmental dimension;
$W_E$: weight factor for the environmental dimension;
F: financial-economical dimension;
$W_F$: weight factor for the financial dimension;
F: political-governance dimension;
$W_P$: weight factor for the political dimension;
S: social-people dimension;
$W_S$: weight factor for the social dimension;

T: technological dimension;

$W_T$: weight factor for the technological dimension;

IC: infrastructure condition;

$W_{ICI}$: weight factor the infrastructure condition.

Equation (4) represents a general, flexible approach that can be adapted according to the indexes and practices that better fit the characteristics, management culture, and level of maturity of the city. For the development of SmartICIPI, a weighted approach can be applied using quantitative and/or qualitative methods to determine the weights. For example, weights can be determined using the Delphi method in combination with more sophisticated techniques such as fair division methods. Fair division methods have been used to solve the problem of dividing goods among several participants. Fair division methods make allocations based on proportionality and envy-freeness. Envy-free methods strive to distribute the resources based on the participants' preferences; the aim is to assign resources to the participant who shows more desire for that resource. Envy-free methods are applied to a variety of divisible and indivisible resources, including transportation funding allocation problems. Chang et al. developed a fair division transportation allocation model (FDTAM) to allocate transportation funds fairly according to individual preferences to prioritize the projects requested for funding; using the own criteria of the participants to set the priorities [69].

It is well recognized by the authors that further research is needed to prepare a step-by-step procedure specifically focused on techniques for the development of smart city indexes for infrastructure management applications.

From a practical perspective, it is observed that the case studies for Lima and Piura in Peru agree with the infrastructure management humanistic approach and 5D conceptual model of smart cities described in this paper.

Table 4 shows the relationship between urban variables and components of the Metropolitan Lima Plan and the 5D smart city model. All urban variables in the Metropolitan Lima Plan are encompassed in the 5D model.

Within the framework of PLANMET 2040 for Metropolitan Lima, projects prioritized for smart city implementation can be incorporated into the 5D model of a smart city. Table 5 shows the criteria and relationship of PLANMET 2040 with the 5D model.

In the case of the city of Piura, every proposed initiative was evaluated using a project-customized ranking tool with 11 criteria. Table 6 shows the relationship between these criteria and the 5D smart city model. The comparison indicates that the project ranking tool mainly focuses on the social-people and political-governance dimensions.

**Table 4.** Relationship of the Metropolitan Lima Plan and the 5D smart city model (adapted from [58]).

| LIMA PLANMET 2040 Component | LIMA PLANMET 20240 Variable | 5D Model | | | | |
|---|---|---|---|---|---|---|
| | | Environmental | Financial Economy | Political Governance | Social People | Technological |
| Demographic | • Concentration of opportunities for labors <br> • Strengthening of capabilities | | | | ✓ | |
| Productive economic | • High informality of urban activities <br> • Urban areas central, intermediate, and peripherals with urban potentials | | ✓ | | | |
| Environment and risk of disasters | • Integration and number of fragile ecosystems and open spaces <br> • Metropolitan environmental management system, deficient and ineffective. | ✓ | | | | |
| Housing | • Housing demand | | ✓ | | ✓ | |
| Metropolitan facilities or amenities | • Deficit and territorial imbalance of metropolitan facilities or amenities, with a high level of vulnerability. | | | ✓ | ✓ | |
| Open spaces and ecological infrastructure | • Ecological infrastructure system (hills, valleys, waterfront, others) | ✓ | | | | |
| Urban infrastructure and services | • Deficit in infrastructure and coverage of urban services. | | | | ✓ | ✓ |

**Table 4.** *Cont.*

| LIMA PLANMET 2040 Component | LIMA PLANMET 20240 Variable | 5D Model | | | | |
|---|---|---|---|---|---|---|
| | | Environmental | Financial Economy | Political Governance | Social People | Technological |
| Urban mobility | • Inadequate metropolitan and interregional passenger and freight mobility system. | | | | ✓ | ✓ |
| Immovable cultural heritage and cultural landscape | • Weak presence of state institutions for the preservation of ancient heritage in the city. | | | ✓ | ✓ | |
| Urban land use and management | • Functionality of the territory, land use, and its mixture, concentration, diversity, and dynamism of urban activities.<br>• Generation and urban regeneration, modern and dynamic areas, attractive for the development of activities, companies, and businesses.<br>• Accelerated urban expansion through the growth of informal and irregular urban settlements in high-risk areas with natural, anthropogenic, and biological hazards. | ✓ | ✓ | | ✓ | ✓ |
| Governance and metropolitan governance | • Inadequate governance, coordination, and participation among the metropolitan government, district governments, surrounding regional governments, and other levels of government, as well as civil society actors.<br>• Inefficient public investment without a monitoring system and data and goal updates. | | | ✓ | ✓ | |

**Table 5.** Comparative analysis of PLANMET 2040 criteria and the 5D smart city model.

| Criteria | 5D Model | | | | |
|---|---|---|---|---|---|
| | Environmental | Financial Economy | Political Governance | Social People | Technological |
| Application of Information and communication technologies (ICT) to provide the citizens with infrastructure with a guarantee of sustainable development and improvement of the quality of life of citizens | | | | ✓ | ✓ |
| Improvement and articulation of the existing metropolitan baseline data to improve the decision-making process in territorial planning (public space, easement areas, etc.) | | | ✓ | ✓ | |
| Promoting water and energy consumption | ✓ | | | | |
| Improving human capital by fostering development, attraction, and nurturing talent | | ✓ | | ✓ | |
| Promoting the Electronic Government of Metropolis with the adequate implementation of ICTs. | | | ✓ | | ✓ |
| Implementation of the Smart City Master Plan in Metropolitan Lima grants a budget of 560,000.00 as a source of financing self-sustainable public–private partnership. | | ✓ | | ✓ | |
| Strategic objective of ensuring the safety of road users and reducing the impact of accidents, missions, and congestion on human life and traffic | ✓ | ✓ | | ✓ | |

**Table 6.** Comparative analysis of Piura project criteria and the 5D smart city model.

| Criteria | 5D Model | | | | |
|---|---|---|---|---|---|
| | Environmental | Financial Economy | Political Governance | Social People | Technological |
| 1. Alignment with the city's objectives: defines whether the project idea is aligned with the city's strategy, allowing for the assurance of political, institutional, and financial support | | ✓ | ✓ | ✓ | |
| 2. Stakeholder engagement: indicates the extent to which stakeholders showed interest in the project idea, based on on-site interactions (interviews, workshops, meetings). | | | ✓ | ✓ | |
| 3. Replicability potential: indicates whether the proposed measure has the potential for replication in other cities, at the state and/or national level, as well as knowledge transfer to a wider audience of stakeholders beyond MGI project partners and stakeholders. | | | ✓ | ✓ | |
| 4. Regulatory constraints: helps determine whether local regulations could pose a significant risk to project implementation. | | | ✓ | | ✓ |
| 5. GHG mitigation potential: signifies the MGI project KPIs; project ideas must meet the predetermined GHG mitigation potential through 2030. | ✓ | | | | ✓ |
| 6. Climate change adaptation potential: means the MGI project KPIs; project ideas must meet the specific climate change adaptation indicators. | ✓ | | | | ✓ |
| 7. Need for financial support from the public sector. | | ✓ | ✓ | ✓ | |
| 8. Likelihood of obtaining public funding in support of the project. | | ✓ | ✓ | ✓ | |
| 9. Interest in the participation of private sector financial support. | | ✓ | ✓ | ✓ | |
| 10. Project approval risk: indicates the complexity of the project approval process through various levels of government agencies, which poses a significant risk to the successful and timely implementation of the project. | | | ✓ | ✓ | |
| 11. The extent of associated resettlement and rehabilitation issues. | | | ✓ | ✓ | |

In the case of Piura, the proposed initiatives were evaluated through the involvement of stakeholders, including government officials, private companies, non-government organization representatives, and citizens. Workshops and festivals were organized to promote citizen involvement, which is the focus of the social-people dimension in the infrastructure management humanistic approach for a smart city. The integrated roadmap with the proposed measures includes ecological and resilience goals in three main sectors: energy, water, and urban planning.

Regarding the European city development approach, some common areas identified as key components in the infrastructure management humanistic approach for smart cities are:

- A shared vision and long-term action plan for a city should be established through the collaboration and participation of the citizens and local authorities from municipalities.
- Leadership should be promoted, and the mayor of the city should be the main advocate to establish an organization with execution capacity and incorporate a transversal vision for project development and implementation.
- Sustainable business models should be developed with returns for all agents involved in the process. In order to promote financing models and ensure the sustainability of the process, it is necessary to involve the private sector and utilize their skills, knowledge, and resources to generate new business models.
- A new management model and strategies should be introduced to build a strong relationship between the local government and companies in the context of a legal framework that promotes private investments.
- An open, standard, and interoperable technological virtual tool should be adopted to support an open system communication platform for interaction and feedback.

While making sure that citizens and government sectors have full access to the open data system, privacy and security should also be ensured. In smart cities, data breaches and cyberattacks are common and pose a direct threat to the personal and private data of citizens. Implementing security methods such as data encryption, multi-factor authentication, and IoT device security is recommended in addition to regulations such as the General Data Protection Regulation (GDPR). The security and privacy of citizens and governments must be enforced for smart cities to be sustainable.

It is the belief of the authors that the infrastructure management humanistic approach described in this paper is only possible with effective communication among local authorities and citizens to synchronize goals and policies with practical solutions to increasing infrastructure needs. It could be argued that this is unfeasible in practice due to the multiple dimensions that are constantly evolving in a smart city. This is partially true since it is correct to affirm that research hypotheses and variables identified in a study may change in the future and the solutions may no longer apply because of new decision contexts.

As a response to this potential criticism, the infrastructure management framework contemplates a dynamic interaction that should continuously capture changes in the five dimensions and proactively anticipate problems. A question arises about how this framework can be adapted to different contexts and new challenges. The answer requires a review of the five dimensions of a smart city as proposed in this paper. Technology has evolved enough to support a virtual platform to sustain the communication needs and support the timely interaction of the participants while addressing the technological dimension of smart cities. Public awareness of environmental concerns has increased while laws, regulations, and engineering solutions seek to cover the environmental dimension. It is also noted that the financial-economic dimension is closely related to the social dimension since funding is required to implement solutions that will affect citizens directly. It is worth remembering that the word economics "comes from the Greek oikonomia (oikos, house, and nomos, law or rule)" [70], and economics is about how to manage the available resources to fulfill the needs of all the people. The political-governance dimension is challenging by its own nature, although it is implied that the ultimate goal of the policies, regulations, and government practices should be to satisfy the needs of the people. Hence, addressing the social dimension of smart cities remains the main challenge that

must be solved when facing infrastructure problems. According to the authors, education plays a major role in overcoming this challenge because, without an educated society, communication with tolerance and respect becomes more difficult, limiting the development, evolution, and sustainability of a smart city.

Finally, the level of smartness of a city is contingent upon political aspects and systems that digitally assist in the management of the civil infrastructure systems in a city with the integration of public opinion in the project selection and funding allocation processes. Smart city governance approaches should focus on the development of an interconnecting communication system to integrate internal government processes with external organizations in the private sector, academia, and general civil society towards building a community-led innovation knowledge network of practice. Local authorities should promote policies to transform traditional government management practices into a new proactive organizational structure that includes smart digital governance services as part of the implementation of the infrastructure management framework described in this paper for smart cities.

## 8. Conclusions

In the context of the UN Sustainable Development Goals, the infrastructure management framework with the 5D smart city conceptual model can be applied to contribute to the development, evolution, and sustainability of smart cities. It is concluded that the main research problem of managing smart cities by using traditional approaches without a holistic infrastructure management vision has been addressed in this research. The research objective of developing an infrastructure management humanistic framework aligned with an integrated multi-dimensional smart city model that emphasizes the role of education was achieved as a result of the blending of qualitative and quantitative research methodologies used to answer the research question. This is a people-centered approach that focuses on the preservation of urban civil infrastructure to enhance the quality of life of the citizens.

The research results are significant since the infrastructure management framework should contribute to expanding knowledge in the academic field for smart city applications. It should also result in a comprehensive set of benefits for the citizens living in a smart city since the ultimate goal is to fulfill the needs of all the citizens while respecting their cultural heritage and promoting socio-economic equity. A summary of the main results of the research follows:

- A people-centered governance approach with an effective infrastructure management proactive approach is necessary for the success of smart city project initiatives. The integration of the citizens' perspective to build a shared vision together with local authorities should be followed by a proactive leadership attitude and business-oriented policies to promote the cooperation of the public and private sectors for the development of civil infrastructure facilities that will contribute to the evolution and sustainability of a smart city.
- The level of maturity of a smart city should be evaluated through the five dimensions model (5D): (1) environmental, (2) financial-economic, (3) political-governance, (4) social-people, and (5) technological. These five dimensions are the main components of the smart city conceptual approach presented in this paper as part of the infrastructure management framework.
- Different tools and methods, such as the Delphi method, can help enhance citizen and expert collaboration to develop and evaluate proposed smart city infrastructure initiatives.
- To motivate citizen involvement in the planning and management of smart city initiatives, mobile apps such as the "Smart Nation App" adopted by the Singapore government are recommended for accessing project infrastructure-related data to ensure public interaction and transparency of the decision-making process.
- Virtual tools such as Mega City 2070 could be used to visualize project initiatives that will lead to green and sustainable civil infrastructure solutions. Nature-based solu-

tions to infrastructure problems are recommended to create a smart and green environment with responsible energy consumption.

- An effective infrastructure management framework is mandatory for project selection and budget allocation to develop and implement short- and long-term smart city plans. Building information modeling-city information modeling (BIM-CIM), digital twin, GIS, and smart IoT devices can be applied to transform conventional infrastructure management approaches into digital governance platforms that allow public participation in the management process.
- A smart collaborative education approach will prepare citizens not only to provide innovative ideas but also to make the best use of them. With the advancement of technology, a city could offer infrastructure facilities to arrange a combination of live and virtual workshops, seminars, and festivals where people can attend and learn about the environment and novel technological initiatives.

As a final conclusion, it is envisioned that an infrastructure management humanistic approach for smart cities should have a positive impact on encompassing urban project initiatives worldwide. Adopting the infrastructure management framework with the 5D smart city model will contribute to well-informed decisions due to the participation of citizens and local authorities. This approach should also aid local governments in improving their planning and management processes and promote the collaboration of the private sector, academia, and citizens.

Research limitations are related to the nature of the problem itself due to the complexity and dynamic interaction among the five smart city dimensions and their integration into infrastructure management practices. Therefore, future research should focus on developing analytical methods and practical tools to analyze these complex interactions and quantify the effects of civil infrastructure initiatives on the socio-economic development of smart cities using a performance-based metric approach supported by technology that fosters human communication with tolerance and respect in an inclusive living community.

**Author Contributions:** C.M.C. conceptualization, methodology, analysis and interpretation, writing, discussion, supervision, review, and editing of original draft preparation; G.T.S. case study of Lima, Peru, Delphi method; T.S.G. main contributor of the role of education section; M.A.V.C. main contributor of the role of technology and case study of Lima, Peru and review of the original draft; S.S. main contributor for the case study in Piura, Peru; S.L.M. literature review, analysis, writing, and draft preparation. All authors have read and agreed to the published version of the manuscript.

**Funding:** The work presented in this paper was supported by the Universidad Ricardo Palma in Lima, Peru as part of the research project titled: "Smart Cities, Sustainable and Resilient". The case study in Piura, Perú was conducted in collaboration with the Universidad de Piura as part of the Morgenstadt Global Smart Cities Initiative (MGI) project funded by the International Climate Initiative (IKI) of the German Federal Ministry for Economic Affairs and Climate Action (BMWK). Partners of the project: University of Stuttgart, Fraunhofer IAO, University of Piura, Municipalidad Provincial de Piura.

**Data Availability Statement:** All data generated or analyzed during this study are included in this published article.

**Acknowledgments:** The authors would like to acknowledge the support of Iván Rodríguez Chávez, President of the Universidad Ricardo Palma, for the research project titled "Sustainable and Resilient Smart Cities". In addition, our special appreciation goes to the International Society for Maintenance and Rehabilitation of Transport Infrastructures (iSMARTI) and the Instituto de Construcción y Gerencia (ICG) for hosting the 5th International Conference on Transportation Infrastructures in August 2022 (V ICTI) in Lima, Peru. The presentations delivered at V ICTI.: "Planning and Development of Smart Cities Using BIM Tools: An Integrated Management Approach" and "Education, Sustainability, and Development of Smart Cities" were the main references in the preparation of this paper.

**Conflicts of Interest:** The authors declare no conflict of interest.

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
