# Peer review of "An Infrastructure Management Humanistic Approach for Smart Cities Development, Evolution, and Sustainabilityâ€"

_infrastructures, doi:10.3390/infrastructures8090127_

Round 1
Reviewer 1 Report
Summary/Contribution: This work developed a holistic, humanistic smart city infrastructure management framework. A five-dimensional model analyzes environmental, economical, political governance, social, and technical variables. The framework was developed using quantitative and qualitative methods and used to Lima and Piura case studies in Peru. The framework aims to improve inhabitants' quality of life by preserving civil infrastructures in smart cities.Comments/Suggestions: 1. The introduction could benefit from a more concise and focused presentation of the main argument and research question. Specifically, the authors could consider revising the research question to make it more specific and actionable.
2. While the introduction provides a good overview of the concept of smart cities, it could benefit from more concrete examples of how infrastructure management practices can contribute to the development, evolution, and sustainability of smart cities. The authors could consider providing more specific examples of infrastructure management practices that have been successful in the context of smart cities.
3. The paper could benefit from a clearer explanation of how the proposed infrastructure management framework integrates the five dimensions of a smart city. Specifically, the authors could consider providing more detail on how each dimension is addressed in the framework and how they are interconnected. 4. The paper could benefit from a clearer explanation of how the mixed research approach combines quantitative and qualitative research procedures to address the research problem. Specifically, the authors could consider providing more detail on how the different techniques (inductive-deductive methods, literature review, participant and non-participant observation, interviews, questionnaires, focus groups, content analysis, smart city measures, and project ranking procedures for infrastructure management) are integrated to provide a comprehensive understanding of the research problem.
5. The authors could also consider providing more detail on how the Delphi method is used in the second phase of the research methodology. Specifically, the authors could explain how the Delphi method is used to collect feedback and analyze data, and how the four stages presented in Figure 1 are implemented in practice.
6. The paper could benefit from more explicit explanations of how the research methodology addresses the multi-dimensional nature of smart cities, and how it integrates humanistic and education perspectives. The authors could consider providing more examples of how the methodology incorporates these perspectives, and how they contribute to the overall research approach.
7. The paper could benefit from a more explicit discussion of the limitations and potential challenges of the proposed approach. Specifically, the authors could consider addressing potential criticisms of the approach and discussing how it could be adapted to different contexts and challenges.
8. The authors may add a paragraph about the security aspects related to smart cities since this point represents a very important aspect to cover. For this purpose, they may consider including the following interesting references:
a. Krichen, Moez, and Roobaea Alroobaea. "A new model-based framework for testing security of iot systems in smart cities using attack trees and price timed automata." 14th international conference on evaluation of novel approaches to software engineering. SCITEPRESS-Science and Technology Publications, 2019. b. Elmaghraby, Adel S., and Michael M. Losavio. "Cyber security challenges in Smart Cities: Safety, security and privacy." Journal of advanced research 5.4 (2014): 491-497.
May be improved
Author Response
The authors genuinely appreciate the comments of the reviewer that have contributed to improving the quality of the paper. We conducted a comprehensive review of the paper to address these comments, expanding the concepts and providing more examples with details as requested by the reviewer. Comments from the reviewers asking for more details and specific examples resulted in the addition of six pages and nineteen references to the previous version.
The reviewer’s comments were beneficial, and we consider that all the comments have been addressed. A detailed response to each of the reviewer’s comments is found in the attached file.

Reviewer 2 Report
This article proposed an infrastructure management humanistic approach with a smart city conceptual model to overcome the shortage of lacking a holistic integrated vision of infrastructure management for smart cities. The research results are significant to help us understand how to build a smart city. However, there are still some details to be clarified or improved.
(1)The research methods involved includes quantitative and qualitative ones. The Delphi method was adopted in this article, but the specific application process was not well integrated with the case study, and the analysis results were not clearly provided. If possible, please provide more details related to the application of the Delphi method.
(2)The most trending smart city indexes recommended for the 5-D humanistic approach are the IMD-SUTD Smart City Index, and IESE Cities in Motion Index. The authors also mentioned that such two indexes cover the five dimensions of the humanistic model. Hence, in terms of the logical organization of the article, the novelty of the proposed 5-D model was weakened greatly. It tends to leave an impression that the proposed 5-D model was just a high-level summary of existing achievements. Correspondingly, for such two existing indexes, the differences between them and the proposed 5-D conceptual model should be clearly pointed out, and their defects and shortcoming should also be given.
Author Response

(The authors gave the same response as above.)

Reviewer 3 Report
Relevant manuscript.
Title:
- in this second version, it is now adequate, with title and subtitle:
An Infrastructure Management Humanistic Approach for Smart Cities Development, Evolution, and Sustainability
Abstract:
- adequate
- ok contextualization/presentation of the topic(s) or theme(s)
- ok research objective was missing
- ok scientific research methodology
- ok description of the results achieved or previous results obtained
- ok conclusion
keywords:
- consistent: civil infrastructure systems; management humanistic approach; smart city; sustainabil- 32
ity; 5-Dimensional model; BIM-CIM, Digital Twin, Peru.
Introduction:
- adequate
- satisfactory from a scientific methodological
- ok contextualization/presentation of the topic(s) or theme(s)
- ok research problems; manuscript objective and research justifications (or relevance)
Literature Review:
- chapters 3, 4 and 5 suitable: enlarged
Research Methodology:
- adequate chapter 2: scientifically expanded
Development and/or Research Results and/or Discussions:
- very good (since the previous version)
6. Case Study: Project to Develop Smart, Sustainable, and Resilient Cities in Peru
- very good (since the previous version)
- please correct numbers 5.1.2. (should be 6, including in two parts of the manuscript) - revise all numbers
7. Discussion
- very good (since the previous version)
8. Conclusion:
- satisfactory from a scientific methodological point of view
- better: “contextualization with Section 1. Introduction
- better: “rescue or closing with research objectives” /
- better: summary of results obtained
- better: scientific contributions
- ok: research limitations (scientific limitations)
- suggestion: it could also include a "final" or "main result of the research for the cities and their respective citizens, where the technological instruments of the Smart City have brought.
- better: the Conclusion with a scientific connotation, as well as, Introduction.
References:
- satisfactory, sufficient, enough: It could still improve with other updated articles/papers from Smart City or Strategic Digital City that contain the strategic vision of the applications of Information Technology resources in cities, thus expanding the quality of life of citizens and also, helping better the management of the mentioned cities in Peru.
Author Response

(The authors gave the same response as above.)

Round 2
Reviewer 1 Report
The authors considered my comments and suggestions
Can be improved